# Enhancement of Lycopene Synthesis via Low-Frequency Alternating Magnetic Field in *Brassica trispora*

Hong Wang [1,†], Jiayang Hou [1,†], Dongxu Wang [1], Maohua Yang [2] and Jinlong Liu [1,*]

1   College of Food Science and Biology, Hebei University of Science and Technology, Shijiazhuang 050018, China; m15232339617@163.com (H.W.); houjy666@126.com (J.H.); eastxuw@163.com (D.W.)
2   CAS Key Laboratory of Green Process and Engineering, Institute of Process Engineering, Chinese Academy of Sciences, Beijing 100190, China; mhyang@ipe.ac.cn
*   Correspondence: jlliu18@126.com
†   These authors contributed equally to this work.

**Abstract:** In recent years, magnetic fields have emerged as a non-thermophysical treatment with a significant impact on microbial fermentation processes. *Brassica trispora* is a microorganism known for its industrial-scale production of lycopene and high yield of single cells. This study aimed to investigate the impact of low-frequency magnetic fields on lycopene synthesis by *Brassica trispora* and elucidate the underlying mechanism for enhancing lycopene yield. The results indicate that both the intensity and duration of the magnetic field treatment influenced the cells. Exposing the cells to a 0.5 mT magnetic field for 48 h on the second day of fermentation resulted in a lycopene yield of 25.36 mg/g, representing a remarkable increase of 244.6% compared to the control group. Transcriptome analysis revealed that the alternating magnetic field significantly upregulated genes related to ROS and the cell membrane structure, leading to a substantial increase in lycopene production. Scanning electron microscopy revealed that the magnetic field treatment resulted in a rough, loose, and wrinkled surface morphology of the mycelium, along with a few micropores, thereby altering the cell membrane permeability to some extent. Moreover, there was a significant increase in intracellular ROS content, cell membrane permeability, key enzyme activity involved in lycopene metabolism, and ROS-related enzyme activity. In conclusion, the alternating frequency magnetic field can activate a self-protective mechanism that enhances lycopene synthesis by modulating intracellular ROS content and the cell membrane structure. These findings not only deepen our understanding of the impact of magnetic fields on microbial growth and metabolism but also provide valuable insights for developing innovative approaches to enhance carotenoid fermentation.

**Keywords:** magnetic field; lycopene; *Brassica trispora*; transcriptome; reactive oxygen species





## 1. Introduction

The Earth harbors a substantial magnetic field. As an environmental factor, the magnetic field (MF) pervades nature and profoundly influences the behavior and physiology of organisms. Recent years have seen extensive research by domestic and international scientists on the biological magnetic effects of artificial magnetic fields on microorganisms, plants, and animals [1]. The manipulability of microorganisms has led to extensive research on the effects of magnetic fields on microbial growth and the synthesis of metabolic product synthesis. According to Zhang's [2] study, the yield of yellow and red pigments significantly increased at an intensity of 0.4 mT in the frequency-modulated magnetic field. However, the increase was smaller and not statistically significant at 0.8 mT and 1.0 mT. Furthermore, the duration of exposure also affected the pigment yield. Magnetic fields have the potential to influence both metabolic products and enzyme activity, by either enhancing or inhibiting the latter. For instance, magnetic fields have

been found to enhance the activity of black fungus oxidoreductase and peroxidase [3]. On the other hand, Aboneima [4] and colleagues discovered that magnetic fields inhibit the activity of carboxymethylcellulosease (CMCase) in black fungi. The decline in enzyme activity starts after 2 h of exposure, and nearly half of the activity is lost after 10 h. The mechanisms by which magnetic fields affect microorganisms include specific and nonspecific effects. Specific effects involve the ability of specific magnetoreceptors in an organism to respond to magnetic fields, leading to gene expression and an increased activity of certain proteins and substance metabolism. This, in turn, enhances the synthesis of specific metabolites. Zhang et al. [5] found that a single gene encoding *Isca1* could act as a magnetic actuator. Additionally, scanning electron microscopy (SEM) can be used to observe the morphology of *Brassica trispora* and the cell membrane [6]. Some studies have also shown that the magnetic field can alter the morphology of *Brassica trispora* and the permeability of the cell membrane. Therefore, this phenomenon is an important criterion for assessing the response of *Brassica trispora* to the magnetic field.

Lycopene, a red natural pigment, serves as an intermediate in the synthesis of various carotenoids and is a product of oxidative stress. It is found in plants, microorganisms, and humans, with the highest concentration occurring in vegetables and fruits such as carrots, tomatoes, watermelons, and pomegranates [7]. Additionally, lycopene can be synthesized through microbial fermentation. Lycopene exhibits antioxidant [8], anticancer [9], cardiovascular disease-prevention [10], and immune-enhancement properties [11], making it widely used in the food, pharmaceutical, health product, and other industries. Research on lycopene has increasingly become a focal point in scientific investigations [12]. However, no studies have been reported on the influence of magnetic fields on the metabolism of *Brassica trispora*, a crucial microorganism in lycopene production. The study aims to examine the impact of various magnetic fields on lycopene production by *Brassica trispora* and preliminarily explore the underlying reasons and mechanisms. This study establishes a theoretical foundation for utilizing magnetic fields to enhance microbial fermentation.

## 2. Materials and Methods

### 2.1. Strain and Culture Condition

*Brassica trispora Fly916* (−) and *Brassica trispora Fly915* (+): the strains were preserved by Hebei fermentation engineering technology research center, Shijiazhuang, China.

Solid culture: strain line inoculation and PDA solid plate, inverted culture at 28 °C for 4 days.

Seed culture: corn starch 40 g/L, soybean cake powder 23 g/L, $KH_2PO_4$ 0.5 g/L, and VB1 0.02 g; pH values were adjusted to 6.3 and sterilized at 121 °C for 20 min under aseptic conditions, the spores were scraped off for 4 days with normal saline, and the concentrations of *Brassica trispora* (−) and *Brassica trispora* (+) spores were controlled to 104/mL and 105/mL. A total of 1 mL of positive and negative spore liquid was inoculated in seed medium and cultured at 28 °C and 180 r/min for 2 days.

Fermentation medium: corn starch 25 g/L, soybean cake powder 50 g/L, soybean oil 40 g/L, $KH_2PO_4$ 1 g/L, and $MgSO_4$ 0.1 g/L; the pH value was adjusted to 6.7 and sterilized at 121 °C for 20 min under aseptic conditions, the positive and negative seed media growing for 2 days were inoculated in the fermentation medium at the proportion of 1:8, the inoculum amount was 10% r/min at 28 °C for 5 days, and 0.2 g gamma L imidazole (cyclase inhibitor) was added at 48 h of fermentation.

### 2.2. Main Reagents and Instruments

Ethy lacetate, magnesium sulfate, corn starch, soybean cake powder, and potassium dihydrogen phosphate were purchased from Tianjin Yongda Chemical Reagent Co., Ltd. (Tianjin, China) Soybean oil, vitamin B1, potato glucose agar, and potato glucose broth were purchased from Beijing Soleibao Technology Co., Ltd. (Beijing, China). The ROS detection kit, sod detection kit, and cat detection kit were purchased from Beijing Suolaibao Technology Co., Ltd. (Beijing, China); the PDS ELISA kit was purchased from Shanghai

Fuyu Biological Co., Ltd. (Shanghai, China). The NEB Next Ultra Directional RNA Library Prep Kit for Illumina was purchased from New England Biolabs Inc., Ltd. (Ipswich, MA, USA) The Prime Script TM first stand cDNA Synthesis Kit was purchased from Bao Nippon Biotechnology Co., Ltd. (Shanghai, China), the spectrophotometer was purchased from Shanghai Yuanxian instrument Co., Ltd. (Shanghai, China), the electrothermal blast drying box was purchased from Shanghai Yiheng Scientific instrument Co., Ltd. (Shanghai, China), the highspeed centrifuge was purchased from Shanghai Anting Science instrument Factory, and the magnetic field and light incubator was purchased from Inductor (Wuxi) Induction Technology Co., Ltd. (Wuxi, China)

### 2.3. Magnetic Field Treatment

The instrument used for fermentation is a magnetic field oscillation incubator produced by INDUC Scientific Co., Ltd. (Shanghai, China) in China. This incubator can be equipped with alternating magnetic fields and can adjust the temperature and shaker speed. The equipment regulates the temperature in the chamber using PID self-tuning control of the heating and cooling system, achieving a temperature accuracy of $\pm0.2$ °C and uniformity of $\pm0.5$ °C. This capability allows for the exclusion of thermal effects caused by the magnetic field and other influencing factors. The equipment has been utilized in scientific research [13,14]. It has an adjustable magnetic field range of 0~5 mT and is classified as a low-frequency magnetic field incubator. Mycelia were isolated by sequentially applying various magnetic field strengths (0, 0.2, 0.5, 0.7, and 1 mT), treatment time points (1, 2, and 3 days), and treatment durations (4, 12, 24, 48, and 72 h) for up to 144 h to the fermentation substrate during the fermentation process. Biomass and lycopene production were measured. Fermentation products without magnetic field treatment were used as the control, and each experiment was repeated three times.

### 2.4. Analytical Method

2.4.1. Detection of Biomass and Lycopene Yield

The strain underwent a fermentation process for 144 h, eventually transitioning into the decline phase. Therefore, the fermentation was carried out under normal conditions until the second day. After that, a magnetic field was applied for 48 h, and the fermentation was switched off until 144 h before determining the lycopene yield and biomass. Mycelial dry weight was determined using the weighing method. Following fermentation, the fermentation broth underwent centrifugation at 8000 rpm for 5 min, followed by three washes with distilled water. Subsequently, it was dried at 80 °C until a constant weight was attained. Lycopene content was determined using the colorimetric method. The centrifuged mycelia were vacuum-dried at 40 °C, ground to disrupt the cell wall, and subsequently extracted with ethyl acetate until becoming colorless [15]. The extracted solution was filtered and diluted, and its absorbance was measured at a wavelength of 502 nm. Lycopene content was quantified using a standard curve (Figure S1) of lycopene [16].

2.4.2. Determination of Process Curve

The lycopene content and biomass of *Brassica trispora* were measured every 24 h throughout the entire fermentation cycle under the optimal magnetic field treatment conditions. Process curves were generated for both the untreated magnetic field conditions and the treated magnetic field conditions.

2.4.3. Determination of Cell Permeability of Mycelium

The relative electrical conductivity of mycelial cells served as an indicator of membrane permeability. The filtered and washed mycelia were dried using filter paper to eliminate moisture. Each sample was standardized for mass and placed in a tightly sealed triangular flask containing 5 mL of distilled water. The extraction was conducted at a constant temperature of 20 °C in an incubator for 24 h with intermittent shaking. The electrical conductivity of the extraction solution was determined using a digital conductivity meter.

Subsequently, the samples were immersed in a boiling water bath for 30 min, cooled to room temperature, and the total electrical conductivity was measured. The relative electrical conductivity (%) = (extract conductivity/total conductivity) × 100.

### 2.4.4. Scanning Electron Microscope (SEM) Observation

The sample-processing method described by Cheng et al. [17] was adopted. A volume of 1 mL of fermentation broth was extracted and transferred to a 2 mL centrifuge tube. The mycelia were then sedimented by centrifugation at 8000 rpm/min, fixed with a 2.5% glutaraldehyde solution, and stored at 4 °C for 24 h. The mycelial precipitate was then washed three times with PBS buffer and subjected to gradient dehydration using ethanol. Two consecutive replacements were performed using ethyl acetate, with each replacement lasting 20 min. Lastly, the samples were dried at room temperature for 2 days, coated with a layer of gold using sputtercoating, and observed using scanning electron microscopy.

### 2.4.5. Transcriptome Analysis

The total RNA of the mycelium samples was extracted and the library was established using the NEB Next Ultra II RNA Library Prep Kit for Illumina (New England Biolabs Inc.; Ipswich, MA, USA). The transcriptional library of the samples was sequenced using the Illumina sequencing platform. The differentially expressed genes were analyzed using DESeq (v1.38.3) software, and the conditions for screening differentially expressed genes were as follows: multiple of the differential expression |log2FoldChange| > 1,significant Pmurvalue valuation < 0.05. Through the functional analysis of the GO and KEGG databases, the differentially expressed genes were annotated and enriched in the metabolic pathway.

### 2.4.6. RT-qPCR

Six related genes were screened for real-time fluorescence quantitative PCR to verify the transcriptome data. The primers used are shown in Table 1. cDNA was synthesized using the Prime Script TM first stand cDNA Synthesis Kit. The RT-qPCR reaction system (20 μL) consisted of 2 × SYBR real-time PCR premixture and 10 μL of upstream- and downstream-specific primers (10 μmol/L), with each 0.4 μL primer free ddH$_2$O, 8 μL. The reaction conditions were 95 °C for 5 min, 95 °C for 15 min, and 60 °C for 30 min, for 40 cycles. With tef1 as the internal reference gene [18], the relative mRNA expression of the target gene was calculated using the 2Ct analysis method [19].

**Table 1.** Primers used for RT-qPCR assays.

| Gene Name | Gene Code | Forward Primer Sequence (5′3′) | Reverse Primer Sequence (5′3′) |
|---|---|---|---|
| *carRA* | gene_8655 | CATCTCGTCGTTGGTTCA | AAGCATAGGCAATAACACAAG |
| *carB* | gene_5071 | GGCACAGATATAACTTGA | TTATTCTTATTGGCTTCCT |
| *sod* | gene_5118 | ATCACTACAATCCTACTG | ACCATACTTCTTCCAATA |
| *cat* | gene_6144 | CTATGCTACCAGAGATATG | CCAGACCTTAGTTACATC |
| *iscA* | gene_10317 | CTGCTGCCAACTCGTTAA | CTGCTGGTGTCAGTGTAAG |
| *elo3* | gene_7519 | TGGTCATCAAGAAGAAGA | GTCAAGTTCAGGATAATAGG |
| *tef1* | | GGTAAGTCTACCACCACTGGTCACT | CAAGAGGAGGGTAGTCAGTGTAAGC |

### 2.4.7. Data Analysis

The experimental data were counted using SPSS 20.0 software (Version 20.0, Chicago, IL, USA). The drawing was made using Origin 2018 software (Origin Lab Corporation, Northampton, MA, USA). All data are expressed as mean ± standard deviation. Each experiment was repeated at least three times in duplicate for statistical analysis.

## 3. Results

### 3.1. Magnetic Field Intensity

Previous studies have demonstrated that magnetic fields exert diverse effects on microorganisms, encompassing magnetic field intensity, treatment timing, and exposure duration [20]. Consequently, the influence of magnetic fields on lycopene production in *Brassica trispora* was examined under the specified magnetic field conditions. The pre-test revealed that a magnetic field strength over 1 mT would affect the normal growth and metabolism of *Brassica trispora* and reduce its lycopene production. Therefore, in this experiment, a magnetic field strength of 0~1 mT was used. Lycopene production was shown after 144 h of fermentation, as depicted in Figure 1. Figure 1 shows a significant increase in lycopene production at magnetic field intensities of 0.2, 0.5, and 0.7 mT compared to the control group. Conversely, lycopene production decreased at a magnetic field intensity of 1 mT, supporting the findings of Jialan et al. [2]. Lycopene production reached 870.2 mg/L at 0.4 mT, representing an 86.5% increase compared to the control group's lycopene yield of 466.4 mg/L. Consequently, the optimal magnetic field intensity for the treatment was identified as 0.5 mT.

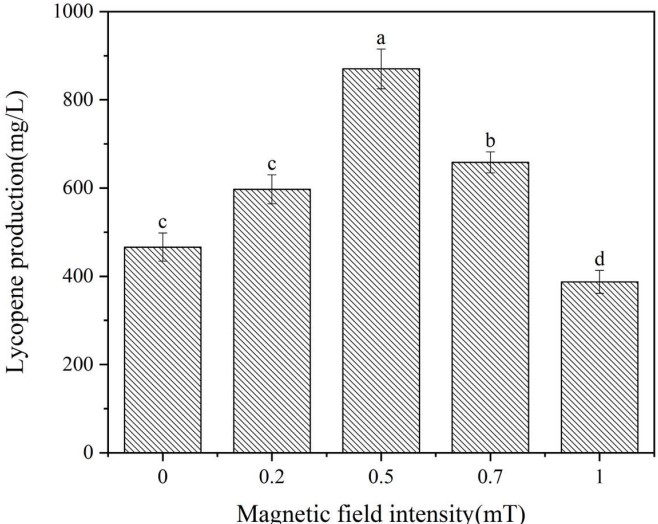

**Figure 1.** Effects of magnetic field intensity on lycopene yield. Different lowercase letters indicate significant differences ($p < 0.05$).

### 3.2. Magnetic Field Processing Time

To fully analyze the optimal magnetic field treatment time for the entire fermentation process, this experiment simultaneously analyzed both the treatment time point and the treatment duration from two aspects. The fermentation broth was exposed to a 0.5 mT magnetic field for different durations (4, 12, 24, 48, and 72 h) on days 1, 2, and 3. Subsequently, the fermentation was continued for 144 h, during which lycopene production and biomass were measured (Table 1). Table 2 reveals a minor, albeit insignificant, inhibition in biomass when compared to the control group. Lycopene production remained nearly unchanged upon magnetic field treatment on the first day of fermentation, while a substantial increase was observed on the second and third days. This can be attributed to the primary production of lycopene during the microorganisms' stable growth phase [16]. The highest lycopene production was achieved after 48 h of exposure, followed by a gradual decline in production with longer treatment durations compared to the control group. This decline in production may be attributed to the prolonged exposure to the magnetic field, which inhibits normal microbial metabolism [21]. On the second day of fermentation, lycopene production reached 25.36 mg/g after 48 h of exposure to a magnetic field, resulting in a 244.6% increase compared to the control group's production of 7.36 mg/g. On the third day of fermentation, lycopene production reached 15.15 mg/L after 48 h of magnetic field

exposure, resulting in a 105.8% increase. Thus, the optimal duration for magnetic field treatment was identified as 48 h of exposure on the second day of fermentation.

**Table 2.** Effects of the magnetic field intervention period and exposure time on biomass and lycopene yield.

| Exposure Period (d) | Exposure Time (h) | Dry Cell Weight (g/L) | Lycopene Yield (mg/g) |
|---|---|---|---|
| Control | 0 | 63.3 ± 3.40 | 7.36 ± 0.38 |
| | 4 | 60.4 ± 4.30 | 6.81 ± 0.31 [e,*] |
| | 12 | 58.3 ± 3.50 | 8.02 ± 0.21 [b] |
| 1 | 24 | 56.1 ± 3.80 | 7.81 ± 0.17 [c] |
| | 48 | 57.2 ± 4.20 | 8.60 ± 0.17 [a] |
| | 72 | 58.3 ± 5.30 | 7.23 ± 0.26 [d] |
| | 4 | 59.3 ± 2.70 | 9.80 ± 0.23 [e] |
| | 12 | 58.1 ± 3.40 | 12.45 ± 0.35 [d] |
| 2 | 24 | 56.3 ± 4.50 | 13.50 ± 0.32 [c] |
| | 48 | 58.1 ± 4.60 | 25.36 ± 0.42 [a] |
| | 72 | 60.2 ± 3.20 | 14.82 ± 0.25 [b] |
| | 4 | 59.3 ± 2.80 | 10.09 ± 0.19 [e] |
| | 24 | 58.5 ± 5.20 | 11.84 ± 0.16 [c] |
| 3 | 48 | 60.2 ± 3.20 | 15.15 ± 0.27 [a] |
| | 72 | 57.1 ± 4.60 | 12.98 ± 0.15 [b] |
| | 12 | 61.2 ± 2.40 | 10.99 ± 0.22 [d] |

[*,a–e] Different lowercase letters in the same column indicate significant differences ($p < 0.05$).

### 3.3. Process Metabolic Curve

Figure 2 illustrates the metabolic profiles of *Brassica trispora* during fermentation under the optimized magnetic field treatment conditions (Table S2). The treatment involved exposing the culture to a 0.5 mT magnetic field for 48 h on the second day of fermentation. Both the control group and the treatment group exhibited similwar patterns in biomass and lycopene production, showing a logarithmic growth phase from 72 to 120 h, followed by subsequent stabilization. The magnetic field had a significant positive effect on lycopene production while also showing a minor inhibitory effect on microbial growth, although with negligible consequences.

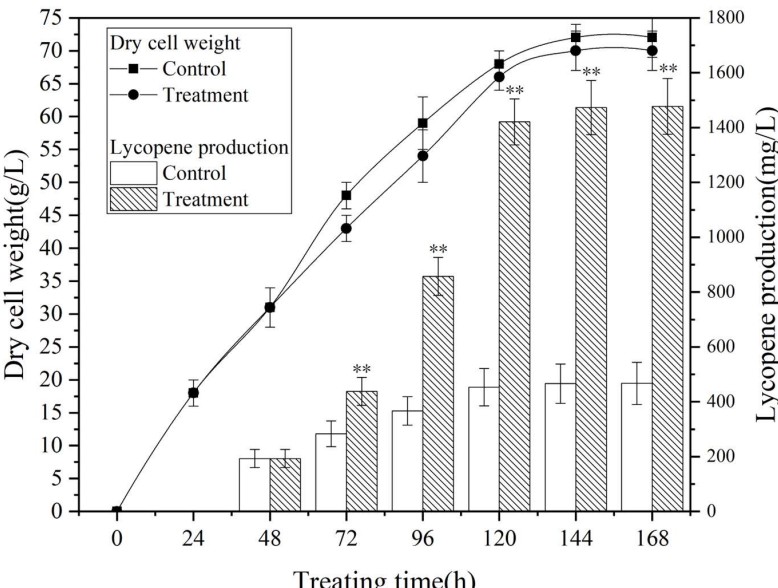

**Figure 2.** Fermentation curve of *Brassica trispora* under the optimal magnetic field conditions (exposure to 0.5 mT magnetic field for 48 h on the second day of fermentation, ** $p < 0.01$).

### 3.4. Transcriptome Analysis of the Effects of Magnetic Field on Bacteriae

### 3.4.1. Differentially Expressed Genes

To examine the impact of magnetic fields on the global transcriptome of Streptomyces brasiliensis during fermentation, it was treated with the optimal magnetic field conditions mentioned earlier and compared with the control group. The mycelium cultures underwent a 144 h fermentation period, and their transcriptomes were subsequently sequenced and analyzed. The results confirmed the reliability of the sequencing by demonstrating an accuracy of base identification exceeding 94%, as indicated by Q30 values. Differential gene expression analysis was performed using DESeq, with the criteria for selecting differentially expressed genes set as |log2FoldChange| > 1 and a significant $p$ value < 0.05. Figure 3 displays the volcano plot, which visualizes the differentially expressed genes between the two conditions. This transcriptome sequencing identified a total of 581 differentially expressed genes, including 405 significantly upregulated genes and 176 significantly downregulated genes.

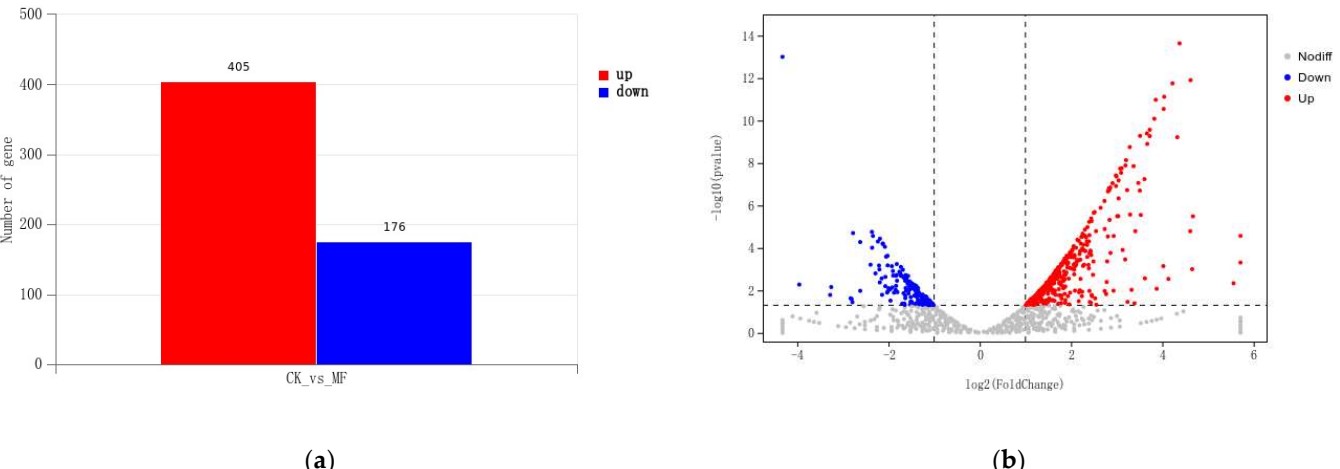

(**a**)                                                                                          (**b**)

**Figure 3.** Comparative transcriptomic analysis of magnetic field and control group. (**a**) Differential expression analysis of the magnetic field group in comparison to the control group, after 120 h of growth. (**b**) Log2 ratio of magnetic field group/control group vs. log10 ($p$ value) in either strain. Red dots represent upregulated genes (log2 Fold Change ≥ 1, and DESeq $P$adj value < 0.05), and blue dots downregulated genes (log2 Fold Change ≤ 1, and DESeq $P$adj value < 0.05). Gray dots represent genes whose changes in expression were not statistically significant.

### 3.4.2. Differentially Expressed Genes

The differentially expressed genes were subjected to gene ontology (GO) enrichment analysis, which categorized them into three main categories: molecular function (MF), biological process (BP), and cellular component (CC). After annotating the differentially expressed genes, the top 20 significantly enriched GO terms were presented in Figure 4. In the molecular function category, the most prominent GO terms were catalytic activity (GO:0003824) and oxidoreductase activity (GO:0016491). The representative GO terms in the biological process category were carbohydrate metabolic process (GO:0005975) and the G protein-coupled receptor signaling pathway (GO:0007186). Significant enrichment of GO terms in the cellular component category was detected in membrane-related locations, such as the integral component of the membrane (GO:0016021), the intrinsic component of the membrane (GO:0031224), and the membrane (GO:0016020).

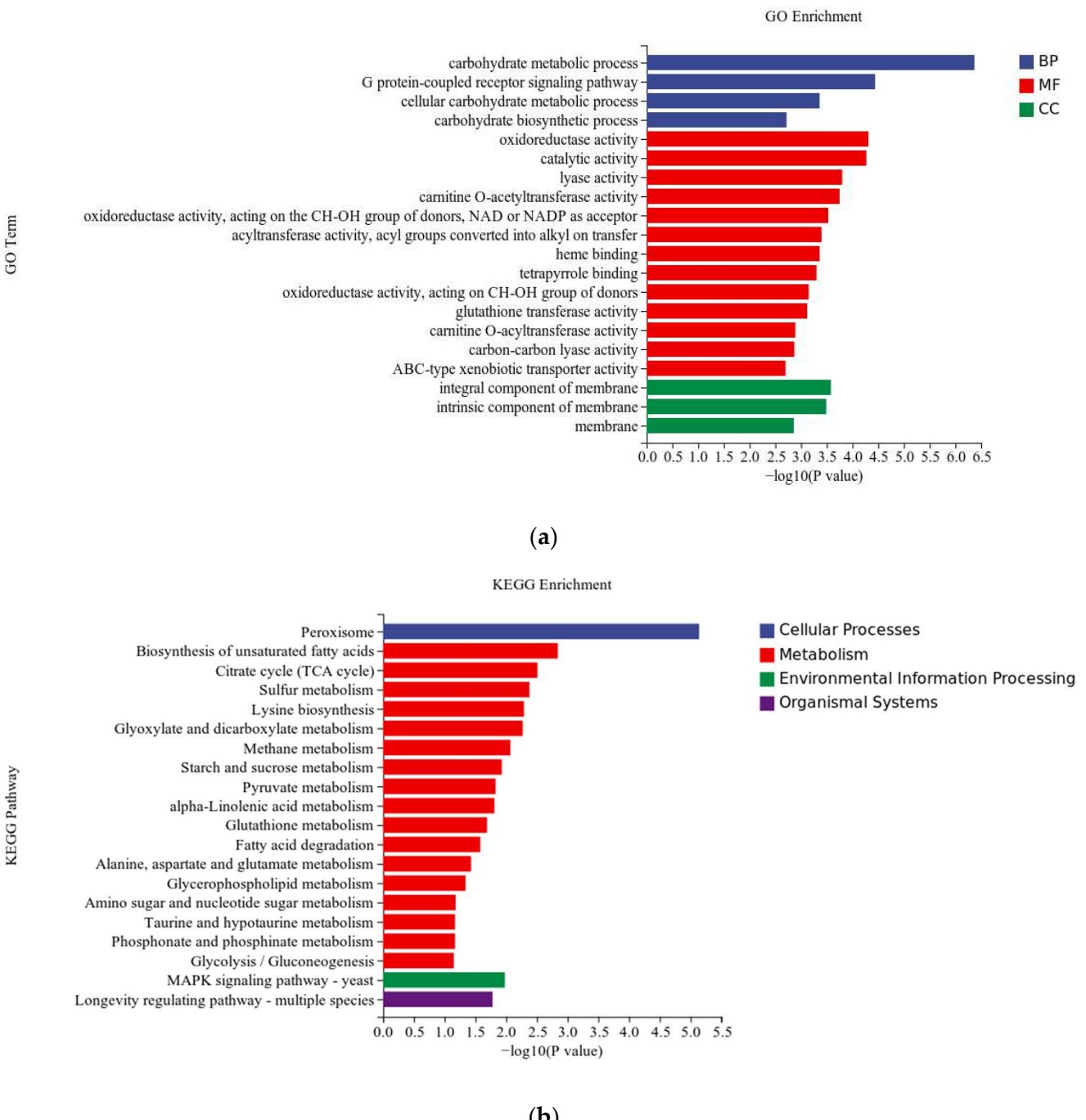

**Figure 4.** GO and KEGG pathway enrichment taxonomic map of differentially expressed genes. (**a**) GO enrichment analysis; (**b**) KEGG enrichment analysis.

Significant enrichment of GO terms in the cellular component category was observed in membrane-related locations, such as the integral component of the membrane (GO:0016021), the intrinsic component of the membrane (GO:0031224), and the membrane (GO:0016020). Figure 4 demonstrates that GO terms related to membrane proteins, such as the G protein-coupled receptor signaling pathway (GO:0007186) and ABC type xenobiotic transporter activity (GO:0008559), were significantly enriched. This suggests that low-frequency magnetic fields may affect transmembrane proteins, leading to increased transcription levels and changes in intracellular calcium and sodium ion concentrations, thereby influencing the cellular metabolism [22]. Other studies have also indicated that low-frequency magnetic fields can alter membrane permeability [23], resulting in changes in ion concentrations both

inside and outside the membrane, and ultimately enhancing the cellular metabolism [24]. The upregulation of cell membrane permeability and membrane protein functionality may be a compensatory response to magnetic fields, safeguarding the cell [25]. From the GO enrichment results, it is evident that the effects of magnetic fields on organisms mainly include oxidative reductive enzymes, membrane-associated functions, and membrane transporters, ultimately impacting cellular growth and metabolism.

Significant enrichment of GO terms in the cellular component category was detected in membrane-related locations, such as the integral component of the membrane (GO:0016021), the intrinsic component of the membrane (GO:0031224), and the membrane (GO:0016020). Figure 4 shows that membrane protein-related GO terms, such as the G protein-coupled receptor signaling pathway (GO:0007186) and ABC type xenobiotic transporter activity (GO:0008559), were significantly enriched. This suggests that low-frequency magnetic fields might impact transmembrane proteins, resulting in elevated transcription levels and modifications in intracellular calcium and sodium ion concentrations, thereby fostering cellular metabolism. Additional studies have shown that low-frequency magnetic fields can modify membrane permeability, leading to changes in ion concentrations inside and outside the membrane, and thereby enhancing cellular metabolism. The upregulation of cell membrane permeability and membrane protein functionality could serve as a compensatory response to magnetic fields, safeguarding the cell.

KEGG pathway enrichment analysis was performed on the differentially expressed genes to investigate the metabolic pathways involved in the response of *Brassica trispora* to magnetic fields. Figure 4 illustrates the top 20 enriched pathways, encompassing cellular processes, metabolism, environmental information processing, and organismal systems. Most of the differentially expressed genes were enriched in cellular engineering and metabolism. The peroxisome pathway was the main enrichment in cellular processes, while metabolism was enriched in the biosynthesis pathway of unsaturated fatty acids, the citric acid cycle (TCA cycle), and the sulfur metabolism pathway. The MAPK signaling pathway was primarily enriched in environmental information processing, and the lifespan regulation pathway was dominant in organismal systems. These enrichment pathways guide further research on the response mechanism of *Brassica trispora* to magnetic fields and validate the results of the GO enrichment analysis.

Among the differentially expressed genes, the redox enzyme metabolism pathway shows significant enrichment. The upregulation of enzymes involved in ROS metabolism, such as superoxide dismutase (gene_8655), catalase (gene_5118), and peroxidase (gene_6907), suggests that magnetic fields can modulate intracellular and extracellular ROS levels. This finding is consistent with the research conducted by Roy et al. [26], which demonstrated that low-frequency magnetic fields can influence biological systems by modulating ROS or free radicals. Moreover, peroxisomes play a crucial role in intracellular ROS production [27], as xanthine oxidase in the peroxisome matrix or membrane can generate $O^{2-}$ [28]. Elevated intracellular ROS levels can damage fungal cells, leading to increased activities of oxidative enzymes like SOD and CAT. Additionally, the cells produce high levels of antioxidants such as lycopene to protect themselves, confirming that magnetic fields may cause sublethal damage to fungal cells and explaining the increased lycopene production [29]. Wang et al. [30] found that ROS can markedly upregulate the transcription of key lycopene metabolic enzymes, namely carotenoid synthase (*carRA*) and lycopene dehydrogenase (*carB*) in *Brassica trispora*, leading to enhanced lycopene production. Importantly, the differentially expressed genes exhibit significant enrichment in the sulfur metabolism pathway, potentially due to the high concentration of magnetic sensor proteins in this pathway [31].

### 3.5. RT-qPCR Analysis

To further analyze the transcriptional activity of the key differentially expressed genes (*carRA*, *carB*, *sod*, *cat*, *iscA*, and *elo3*), their expression levels were measured via the real-time quantitative PCR (RT-qPCR) technique. Lycopene synthesis in *Brassica trispora*

predominantly occurs through the mevalonate (MVA) pathway, where *carRA* and *carB* serve as key enzymes for lycopene synthase and lycopene cyclase [32]. Therefore, carRA and carB were analyzed as important genes. Additionally, Figure 5 illustrates the impact of the magnetic field on the key enzyme genes involved in lycopene metabolism. To validate the increased intracellular oxidative levels, superoxide dismutase and catalase (*sod*, *cat*), enzymes involved in the metabolism of reactive oxygen species (ROS), were selected for analysis. The transcriptomics was validated by analyzing the cell membrane-related gene elo3 results. This upregulation was confirmed through RT-qPCR to validate the transcriptomic results. Notably, Qin et al. [31] discovered that the *iscA* gene, encoding the ironsulfur cluster assembly protein, functions as a biomagnetic sensor in response to magnetic fields. Thus, the effects of magnetic fields on fungal organisms were investigated by analyzing the *iscA* gene. The RT-qPCR results showed consistency with the RNA-Seq data, displaying similar trends and confirming the reliability of the transcriptomic findings in this study.

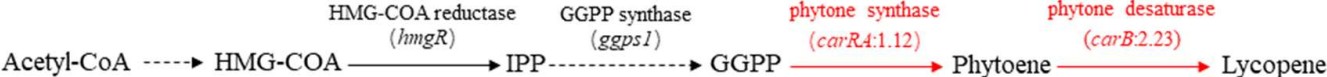

**Figure 5.** Effect of magnetic field on the transcriptional level of genes encoded by key enzymes in the lycopene metabolic pathway.(The black dashed lines represent intermediate processes where multiple actions have been omitted, and the black solid lines represent direct actions.)

The RT-qPCR results, presented in Table 3, indicate substantial upregulation of key enzymes involved in lycopene metabolism (*carRA*, *carB*), ROS-related genes (*sod*, *cat*), and the cell membrane-related gene *elo3*. This further supports the notion that the magnetic field treatment enhances lycopene production by affecting ROS levels and cell membrane integrity. During the magnetic field treatment, the expression of the *iscA* gene, which encodes the ironsulfur cluster assembly protein, was upregulated, potentially because it acts as a magnetic receptor protein (*MagR*) responsive to magnetic fields [31]. *IscA* (*MagR*) can induce membrane depolarization and action potentials through external magnetic field exposure, resulting in intracellular calcium influx and the subsequent activation of the organism's magnetic field response [5].

**Table 3.** Results of RT-qPCR verification of some differentially expressed genes.

| Name | Gene ID | Description | Fold Change of RNASeq | $2^{\Delta\Delta Ct}$ of RT-qPCR |
|------|---------|-------------|------------------------|----------------------------------|
| *carRA* | gene_8655 | Phytoene synthase | 1.12 | 2.15 |
| *carB* | gene_5071 | Phytoene desaturase | 2.23 | 1.47 |
| *sod* | gene_5118 | Superoxide dismutase | 2.42 | 2.15 |
| *cat* | gene_6144 | catalase | 8.18 | 7.80 |
| *iscA* | gene_10317 | Ironsulfur cluster assembly protein | 1.27 | 1.08 |
| *elo3* | gene_7519 | ELO family | 4.26 | 3.19 |

*3.6. Mycelial Morphologye*

The influence of the magnetic field on the microstructure of mycelia of *Brassica trispora* was analyzed using SEM. Figure 4a illustrates the mycelial surface, which appeared smooth, flat, and exhibited minimal bending. The mycelial surface in the magnetic field group (Figure 6b) displayed roughness, looseness, increased folding, and a few micropores. This can be attributed to the elevated levels of intracellular ROS, which induce changes in the structure and permeability of the cell membrane. This finding aligns with the research conducted by Guo Li et al. [33], which documented increased folding on the surface of Grifola frondosa mycelia and an overall more porous mycelial structure. Likewise, Voychuk et al. [34] observed an increase in cell wall invaginations in Saccharomyces cerevisiae yeast cells exposed to magnetic fields,

resulting in modifications to internal metabolic synthesis processes. This could be attributed to the ability of magnetic fields to induce alterations in the local structure of the cell membrane, thereby facilitating the transport of ions and molecules [35]. Microscopic examination of the mycelial color, as depicted in Figure 6c,d, demonstrated that the mycelia in the magnetic field-treated group displayed a more intense red color in comparison to the control group. This further supports the notion that magnetic fields enhance lycopene synthesis.

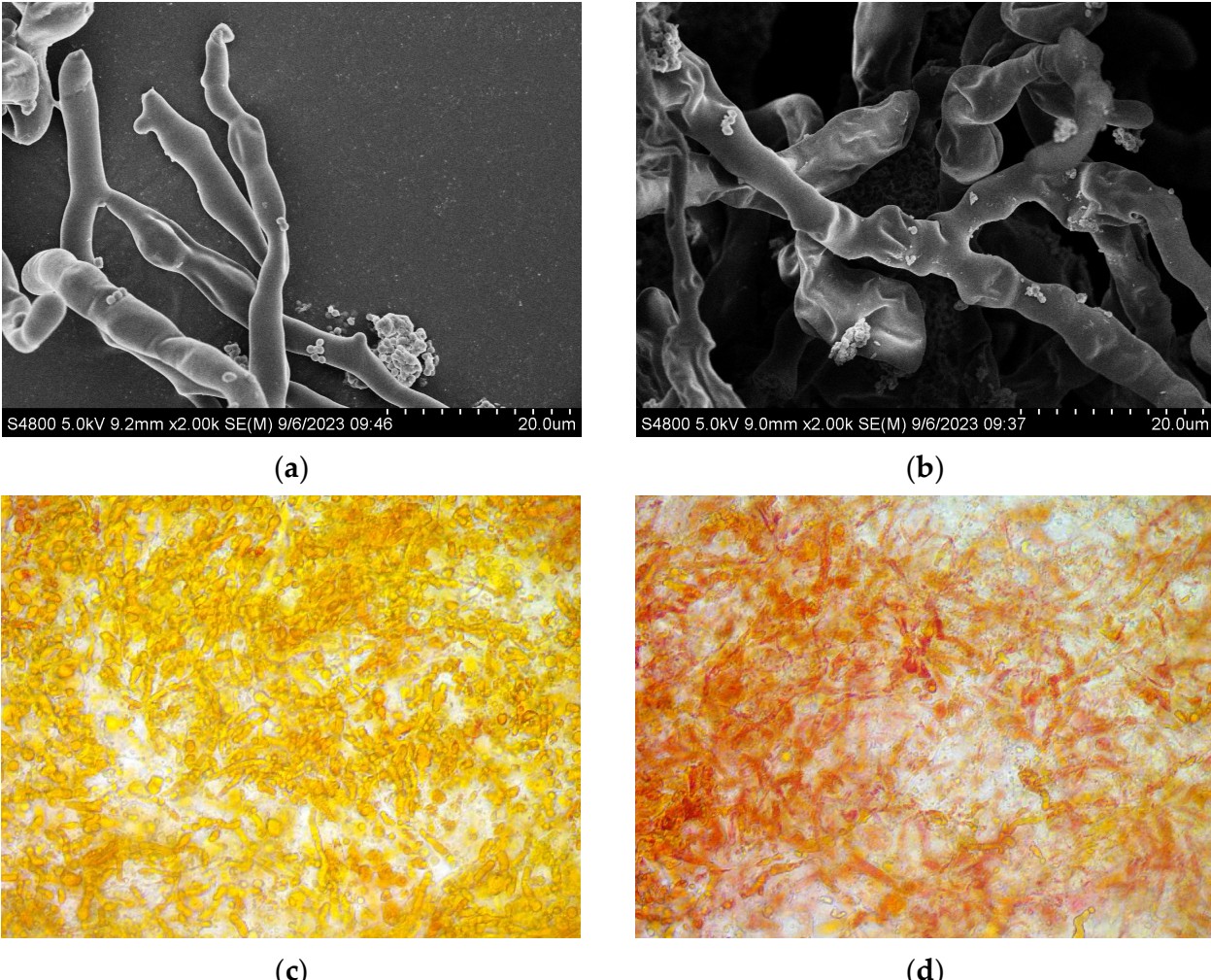

**Figure 6.** Microstructure of *Brassica trispora* under scanning electron microscope (SEM) and microscope. (**a**,**c**) Blank group; (**b**,**d**) magnetic field treatment for 48 h.

*3.7. Effects of Magnetic Field Treatment on Intracellular Reactive Oxygen Species, Cell Membrane Permeabilit, y and Activities of Key enzymes*

To further validate the transcriptome analysis results, as shown in Figure 7, the activities of key enzymes, cell membrane permeability, and intracellular ROS levels were measured. The magnetic field treatment significantly increased intracellular ROS content compared to the control group. Additionally, the activities of ROS metabolism-related enzymes, such as superoxide dismutase (SOD), catalase (CAT), and phytoene desaturase (PDS), the key enzyme in lycopene metabolism, were significantly enhanced (Tables S3 and S4). These findings suggest that the elevated intracellular ROS levels stimulate lycopene production, contributing to the maintenance of intracellular redox balance. Quiles Rosillo et al. [36] made similar observations, reporting that ROS markedly upregulated the transcription levels of *carB* and *carRA*, resulting in enhanced carotenoid accumulation. Wang et al. [30] also demonstrated that the supplementation of $H_2O_2$

increased lycopene production in *Brassica trispora*. These findings align with the study conducted by Qian et al. [37], which revealed that low-frequency alternating magnetic fields can enhance *Monascus* pigment synthesis by modulating intracellular ROS levels.

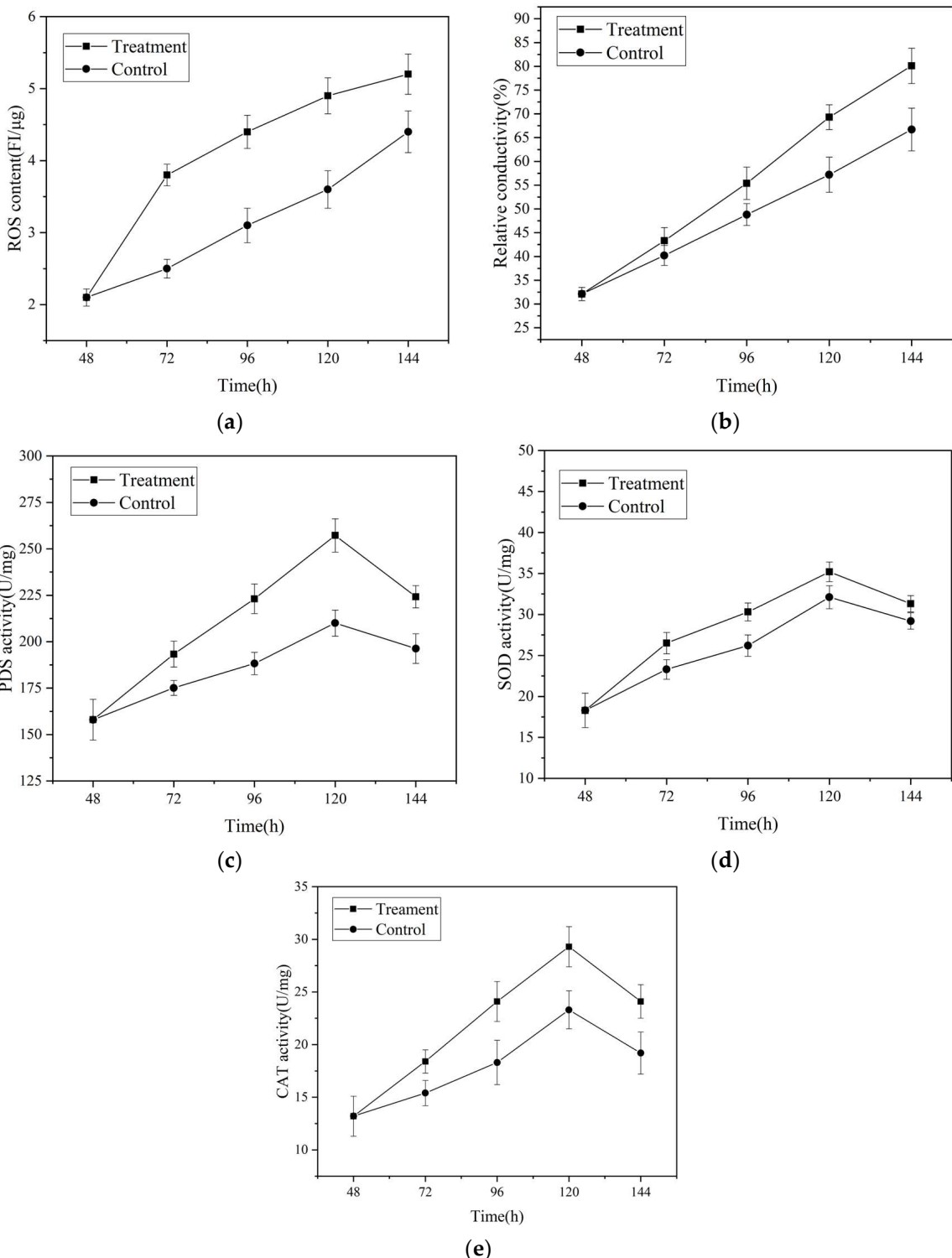

**Figure 7.** The effects of magnetic fields on crucial enzymes, ROS-related enzyme activity, cellular membrane permeability, and intracellular ROS in the lycopene synthesis pathway of *Brassica trispora*. (**a**) Intracellular ROS content; (**b**) cell membrane permeability; (**c**) PDS activity; (**d**) SOD activity; (**e**) CAT activity.

In Figure 7b, a significant increase in cell membrane permeability of the mycelia under the magnetic field treatment is demonstrated when compared to the control group. This implies that magnetic field treatment has the ability to modify cell membrane permeability, ultimately enhancing cellular metabolism [38]. Similarly, Liu et al. [20] discovered that magnetic field stimulation induced changes in the membrane fluidity and permeability of Ganoderma lucidum mycelia, influencing the exchange of substances between the intracellular and extracellular environments and promoting cellular metabolism. Collectively, these findings indicate that magnetic field treatment primarily amplifies intracellular ROS levels(Table S5) and cell membrane permeability(Table S6), providing further confirmation of the results obtained from transcriptome analysis and scanning electron microscopy.

## 4. Conclusions

Overall, this study investigated the effects of magnetic field treatment on lycopene production by *Brassica trispora*. The application of a 0.5 mT magnetic field during the second day of fermentation and a 48 h treatment period resulted in a significant increase in lycopene yield. To understand the underlying mechanisms, scanning electron microscopy, transcriptomics, and real-time quantitative PCR analyses were conducted. The transcriptomic analysis revealed upregulation of genes associated with oxidation-reduction enzymes and membrane-related processes. The reliability of the transcriptomic data was confirmed through scanning electron microscopy and real-time quantitative PCR results. Additionally, the study found that magnetic field treatment led to a significant increase in intracellular ROS content, as well as the activities of SOD and CAT enzymes. These findings suggest that the magnetic field treatment significantly increased the intracellular ROS content and cell membrane permeability, so that the mycelium produced large amounts of the antioxidant lycopene to protect itself. Furthermore, the study identified the expression of the ironsulfur cluster assembly protein (gene_10317) in response to the magnetic field, although its role as a magnetic sensor and the underlying response mechanism remain unclear and require further investigation. Overall, these results offer valuable insights and methods for the carotenoid fermentation industry, and serve as a reference for future studies on magnetoreceptors in *Brassica trispora*.

**Supplementary Materials:** The following supporting information can be downloaded at: https://www.mdpi.com/article/10.3390/fermentation10010069/s1, Figure S1: Lycopene standard curve; Table S1: Effect of magnetic field intensity on lycopene yield; Table S2: Fermentation curve of *Brassica trispora* under the optimal magnetic field conditions; Table S3: The effect of magnetic fields on ROS-related enzyme activity in the lycopene synthesis pathway of *Brassica trispora*; Table S4: The effect of magnetic fields on crucial enzymes activity in the lycopene synthesis pathway of *Brassica trispora*; Table S5: The effect of magnetic fields on intracellular ROS in the lycopene synthesis pathway of *Brassica trispora*; Table S6: The effect of magnetic fields on cellular membrane permeability in the lycopene synthesis pathway of *Brassica trispora*.

**Author Contributions:** J.H. and H.W. contributed equally to this paper; J.H. and H.W. performed the main experiments and prepared figures and tables; J.L. contributed the experiment materials and performed some experiments; J.H., H.W. and D.W. wrote the manuscript text; J.L. and M.Y. designed the experiments. All authors reviewed the manuscript. All authors have read and agreed to the published version of the manuscript.

**Funding:** This research was funded by the Natural Science Foundation of Hebei Province (Grant No. C2021208019).

**Institutional Review Board Statement:** Not applicable.

**Informed Consent Statement:** Not applicable.

**Data Availability Statement:** Data are contained within the article and supplementary materials.

**Acknowledgments:** The authors gratefully acknowledge Jinlong Liu from the Hebei University of Science and Technology (Shijiazhang, China) for providing suggestions for this experimental study, and gratefully acknowledge Maohua Yang from the Institute of process engineering, the Chinese Academy of Sciences (Beijing, China) for providing suggestions for this experimental study.

**Conflicts of Interest:** The authors declare no conflicts of interest.

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
