# Peer review of "Enhancement of Lycopene Synthesis via Low-Frequency Alternating Magnetic Field in Brassica trispora"

_fermentation, doi:10.3390/fermentation10010069_

Round 1

Reviewer 1 Report

Comments and Suggestions for Authors

The article tried to improve the lycopene production from Brassica trispora utilizing magnetic field treatment. The finding provides a high value in industrial application.  However, there are still some parts that authors should work on to improve this article.

1.        The possible mechanism of magnetic field treatment for specific product production improvement from microorganism should be provided in the introduction.

2.        In the 2.1, the information of used microorganism should be corrected. Please provide the explanation of line 72 “B. trispora Fly-916 (-) and B. trispora Fly-915 (+)”. And how to obtain the microorganism?

3.        The title of subchapter “2.3. Training method” should be checked. It is the culture condition. I suggested that the 2.1 and 2.3 should be combined.

4.        In the fermentation process, the strain was culture for 144 hr, but the strain just treated under magnetic field for 72 hr, it that mean the strain has 72 hr rest? The description of 2.4 should be improved, and it is difficult to be understanding.

5.        In the table 2, the productivity (the production of lycopene per cell) should be provided and discussed.  

6.        The active vice should be avoid such as line 278, and please check in all the text.

7.        In the KEGG data, the enhancement of magnetic field treatment for lycopene production seems related to the sublethal damage effect of microorganism? Please check this and give some discussions.

Comments on the Quality of English Language

no comment

Author Response

We thank the reviewers for their suggestions, the parts of the article that have been revised have been marked, and the following are responses to specific suggestions:
1. Deepen the introduction to magnetic fields in the introduction by providing possible mechanisms for the effects of magnetic field treatment on microorganisms.
2. Explained how microorganisms are obtained.
3. Have expressed 2.1 and 2.3 together.
4. Optimized the description of 2.4 to make the experimental method clearer and easier to understand.
5. Modified Table 2 to provide the productivity (the production of lycopene per cell).
6. Modified the active voice in the article.
7. Explained more deeply the relationship between lycopene production by magnetic field treatment and sublethal damage to microorganisms in the KEGG section.

Reviewer 2 Report

Comments and Suggestions for Authors

This is a potentially interesting study that is similar to previously completed work, with additional insight. However, there are major flaws in how the experiments are described, presented, and flow of the document that must be corrected before it can even be properly assessed.

Introduction

No background on why SEM was used, was morphology differences expected?

Major issues

Absolutely no methodology on how the magnetic fields were applied to the microorganisms, or why these levels were chosen other than a single reference. This is critical as it is the entire point of the experiment, and readers must be able to assess how the experiment was conducted, and that the authors are attributing the effects of the experiment correctly. Additionally, all methodology is vague and written in a manner that could not be replicated, be specific when describing replications and experimental design, especially when describing how magnetic fields were applied. This is doubly important when attempting to investigate theoretical mechanisms

Without knowing how the experiments were conducted, it is impossible to evaluate the results. These are potentially interesting results, but could be attributed to other phenomenon such as temperature effects of the equipment depending upon how the experiments completed.

Is the SEM images representative of all the organisms? The cells behind the rough structure appear smooth and similar to the image in (A). Additional images would be helpful if available.

Conclusion:

only tangible conclusions are similar to cited work "Effect of low-frequency magnetic field on formation of pigments of Monascus purpureus" Novel conclusions are vague e.g "These findings offer theoretical insights into carotenoid fermentation under magnetic field conditions" be specific in describing the contributions of this work. The fact that the last section of the conclusions is attributed to another author (reference 34) is very telling, the conclusions need to be completely rewritten to be stating the findings (conclusions) of THIS study.

Comments on the Quality of English Language

Odd hyphenation is included throughout the document in incorrect locations. This must be corrected as it is extremely distracting and indicative of a poorly proofread document. 

Author Response

We thank the reviewers for their suggestions, the parts of the article that have been revised have been marked, and the following are responses to specific suggestions:
1. Improved the introduction, described why SEM is used
2. Main issues: Modified the magnetic field processing method, improved the description of the magnetic field processing method, and introduced the details of the magnetic field processing instrument, which is easy to follow and understand.
3. Provided SEM images with more obvious contrast, which is easier to observe and understand.
4. Rewrote the conclusion to make it more obvious and specific.
